# Answer Set Programming for Regular Inference

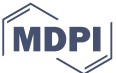

**Wojciech Wieczorek** [1,*], **Tomasz Jastrzab** [2] **and Olgierd Unold** [3]

1    Department of Computer Science and Automatics, University of Bielsko-Biala, Willowa 2,
     43-309 Bielsko-Biala, Poland
2    Department of Algorithmics and Software, Silesian University of Technology, Akademicka 16,
     44-100 Gliwice, Poland; tomasz.jastrzab@polsl.pl
3    Department of Computer Engineering, Wroclaw University of Science and Technology,
     Wyb. Wyspianskiego 27, 50-370 Wroclaw, Poland; olgierd.unold@pwr.edu.pl
*    Correspondence: wwieczorek@ath.bielsko.pl

**Abstract:** We propose an approach to non-deterministic finite automaton (NFA) inductive synthesis that is based on answer set programming (ASP) solvers. To that end, we explain how an NFA and its response to input samples can be encoded as rules in a logic program. We then ask an ASP solver to find an answer set for the program, which we use to extract the automaton of the required size. We conduct a series of experiments on some benchmark sets, using the implementation of our approach. The results show that our method outperforms, in terms of CPU time, a SAT approach and other exact algorithms on all benchmarks.

**Keywords:** answer set programming; non-deterministic automata induction; grammatical inference

## 1. Introduction

The main problem investigated in this paper is as follows. Given a finite alphabet $\Sigma$, two finite subsets $S_+, S_- \subseteq \Sigma^*$, and an integer $k > 0$, find a $k$-state NFA $A$ that recognizes a language $L \subseteq \Sigma^*$ such that $S_+ \subseteq L$ and $S_- \subseteq \Sigma^* - L$. In other words, we are dealing with the process of learning a finite state machine based on a set of labeled strings, thus building a model reflecting the characteristics of the observations. Machine learning of automata and grammars has a wide range of applications in such fields as syntactic pattern recognition, computational biology, systems modeling, natural language acquisition, and knowledge discovery (see [1–5]).

It is well known that NFA or regular expression minimization is computationally hard: it is PSPACE-complete [6]. Moreover, even if we specify the regular language by a deterministic finite automaton (DFA), the problem remains PSPACE-complete [7]. Angluin [8] showed that there is no polynomial-time algorithm for finding the shortest compatible regular expression for arbitrary given data (if P ≠ NP). Thus we conjecture that the complexity of inferring a minimal-size NFA that matches a labeled set of input strings is probably exponential.

For the deterministic case, the problem is NP-complete [9]. Besides, in contrast to the NFAs, for a given regular language there is always exactly one minimum-size DFA (i.e., there is no other non-isomorphic DFA with the same minimal number of states). Therefore, is NFA induction harder than DFA induction? To answer this, let us compare the problem search space sizes expressed by the number of automata with a fixed number of states. Let $c$ be the size of the alphabet and $k$ the number of automaton states. The number of pairwise non-isomorphic minimal $k$-state DFAs over a $c$-letter alphabet is of order $k2^{k-1}k^{(c-1)k}$. The number of NFAs such that every state is reachable from the start state is of order $2^{ck^2}$ [10]. Thus, switching from determinism to non-determinism increases the search space enormously. However, on the other hand, it is well known that NFAs are more compact. A DFA could even be exponentially larger than a corresponding NFA for a given language.

The purpose of the present proposal is twofold. The first objective is to devise an algorithm for the smallest non-deterministic automaton problem. It entails preparing logical rules (this set of rules will be called an AnsProlog program) before starting the searching process. The second objective is to show how the ASP solvers help to tackle the regular inference problem for large-size instances and to compare our approach with the existing ones. Particularly, we will refer to the following exact NFA identification methods [11]:

1. A randomized algorithm using Parallelization Scheme 1 (RA-PS1), with *deg*, *mmex* and *mmcex* variable ordering methods chosen at random,
2. A randomized algorithm using Parallelization Scheme 2 (RA-PS2), with final state combinations chosen at random,
3. An ordered algorithm using Parallelization Scheme 1 (OA-PS1), with *deg*, *mmex* and *mmcex* variable ordering methods chosen according to the given order in a round robin fashion,
4. An ordered algorithm using Parallelization Scheme 2 (OA-PS2), with final state combinations ordered by the number of final states.

We will also refer to a SAT encoding given in [5]. All four above-mentioned methods and a SAT encoding are thoroughly described in Section 4.2. To enable comparisons with other methods in the future, the Python implementation of our approach is made available via GitHub. The Python scripting language is used only for generating the appropriate AnsProlog facts and running Clingo, an ASP solver.

Another line of research concerns the induction of DFAs. The original idea of SAT encoding in this context comes from the work made by Heule and Verwer [12]. Their work, in turn, was based on the idea of transformation from DFA identification into graph coloring, which was proposed by Coste and Nicolas [13]. Zakirzyanov et al. [14] proposed BFS-based symmetry breaking predicates, instead of the original max-clique predicates, which improved the translation-to-SAT technique. The improvement was demonstrated with the experiments on randomly generated input data. The core idea is as follows. Consider a graph $G$, the vertices of which are the states of an initial automaton and there are edges between vertices that cannot be merged. Finding minimum-size DFA is equivalent to a graph coloring with a minimum number of colors. The graph coloring constraints, on the other hand, can be efficiently encoded into SAT according to Walsh [15].

In a more recent approach, satisfiability modulo theories (SMT) are explored. Suppose that $A = (\Sigma, Q = \{0, 1, \ldots, K-1\}, s = 0, F, \delta)$ is a target automaton and $P$ is the set of all prefixes of $S_+ \cup S_-$. An SMT encoding proposed by Smetsers et al. [16] uses four functions: $\delta \colon Q \times \Sigma \to Q$, $m \colon P \to Q$, $\lambda^A \colon Q \to \{\bot, \top\}$, $\lambda^T \colon S_+ \cup S_- \to \{\bot, \top\}$, where $\{\bot, \top\}$ represents logical $\{\text{false}, \text{true}\}$, and the following five constraints:

$$m(\varepsilon) = 0,$$

$$x \in S_+ \iff \lambda^T(x) = \top,$$

$$\forall xa \in P \colon x \in \Sigma^*, a \in \Sigma \quad \delta(m(x), a) = m(xa),$$

$$\forall x \in S_+ \cup S_- \quad \lambda^A(m(x)) = \lambda^T(x),$$

$$\forall q \in Q \quad \forall a \in \Sigma \quad \bigvee_{r \in Q} \delta(q, a) = r.$$

They implemented the encodings using Z3Py, the Python front-end of an efficient SMT solver Z3.

This paper is organized into five sections. In Section 2, we present necessary definitions and facts originating from automata, formal languages, and declarative problem-solving. Section 3 describes our inference algorithm based on solving an AnsProlog program. Section 4 shows the experimental results of our approach and describes in detail all reference methods. Concluding comments are made in Section 5.

## 2. Preliminaries

We assume the reader to be familiar with basic regular language and automata theory, for example, from [17], so that we introduce only some notations and notions used later in the paper.

### 2.1. Words and Languages

An *alphabet* $\Sigma$ is a finite, non-empty set of symbols. A *word* $w$ is a finite sequence of symbols chosen from an alphabet. The length of word $w$ is denoted by $|w|$. The *empty word* $\varepsilon$ is the word with zero length. Let $x$ and $y$ be words. Then $xy$ denotes the *concatenation* of $x$ and $y$, that is, the word formed by making a copy of $x$ and following it by a copy of $y$. As usual, $\Sigma^*$ denotes the set of words over $\Sigma$. A word $w$ is called a *prefix* of a word $u$ if there is a word $x$ such that $u = wx$. It is a *proper* prefix if $x \neq \varepsilon$. A set of words taken from some $\Sigma^*$, where $\Sigma$ is a particular alphabet, is called a *language*.

A *sample* $S$ is an ordered pair $S = (S_+, S_-)$ where $S_+$, $S_-$ are finite languages with an empty intersection (i.e., having no common word). $S_+$ is called the *positive part of* $S$ (*examples*), and $S_-$ the *negative part of* $S$ (*counter-examples*).

### 2.2. Non-Deterministic Finite Automata

A *non-deterministic finite automaton* (NFA) is a five-tuple $A = (\Sigma, Q, s, F, \delta)$ where $\Sigma$ is an alphabet, $Q$ is a finite set of states, $s \in Q$ is the initial state, $F \subseteq Q$ is a set of final states, and $\delta$ is a relation from $Q \times \Sigma$ to $Q$. Members of $\delta$ are called *transitions*. A transition $((q, a), r) \in \delta$ with $q, r \in Q$ and $a \in \Sigma$, is usually written as $r \in \delta(q, a)$. Relation $\delta$ specifies the moves: the meaning of $r \in \delta(q, a)$ is that automaton $A$ in the current state $q$ reads $a$ and can move to state $r$. If for given $q$ and $a$ there is no such $r$ that $((q, a), r) \in \delta$, the automaton stops and we can assume it enters the rejecting state. Moving into a state that is not final is also regarded as rejecting but it may be just an intermediate state.

It is convenient to define $\bar{\delta}$ as a relation from $Q \times \Sigma^*$ to $Q$ by the following recursion: $((q, ya), r) \in \bar{\delta}$ if $((q, y), p) \in \bar{\delta}$ and $((p, a), r) \in \delta$, where $a \in \Sigma$, $y \in \Sigma^*$, and requiring $((t, \varepsilon), t) \in \bar{\delta}$ for every state $t \in Q$. The *language accepted* by an automaton $A$ is then

$$L(A) = \{x \in \Sigma^* \mid \text{there is } q \in F \text{ such that } ((s, x), q) \in \bar{\delta}\}. \tag{1}$$

Two automata are *equivalent* if they accept the same language.

Let $A = (\Sigma, Q, s, F, \delta)$ be an NFA. Then we will say that $x \in \Sigma^*$ is: (a) *recognized by accepting* (or *accepted*) if there is $q \in F$ such that $((s, x), q) \in \bar{\delta}$, (b) *recognized by rejecting* if there is $q \in Q - F$ such that $((s, x), q) \in \bar{\delta}$, and (c) *rejected* if it is not accepted.

### 2.3. Answer Set Programming

Let us shortly introduce the idea of answer set programming (ASP). The readers interested in the details of ASP, alternative definitions, and the formal specification of AnsProlog are referred to handbooks [18–20].

Let $\mathcal{A}$ be a set of *atoms*. A *rule* is of the form:

$$a \leftarrow b_1, \ldots, b_k, \sim c_1, \ldots, \sim c_m. \tag{2}$$

where $a$, $b_i$s, and $c_i$s are atoms and $k, m \geq 0$. The *head* of the rule, $a$, may be absent. The part on the right of '$\leftarrow$' is called the *body* of the rule. The symbol $\sim$ is called *default negation* and, by analogy to database systems, in logic programming it refers to the absence of information. Informally, $a \leftarrow \ldots \sim b$ means: if $\ldots$ and there is no evidence for $b$ then $a$ should be included into a solution. A *program* $\Pi$ is a finite set of rules.

Let $R$ be the set of rules of the form:

$$a \leftarrow b_1, \ldots, b_k. \tag{3}$$

and $\mathcal{A}$ be a set of atoms occurring in $R$. The *model* of a set $R$ of rules without negated atoms is a subset $M \subseteq \mathcal{A}$ which fulfills the following conditions:

1.  if $a \leftarrow .$ is in $R$, then $a \in M$;
2.  if $\leftarrow b_1, \ldots, b_k.$ is in $R$, then at least one $b_i$ $(1 \leq i \leq k)$ is in $\mathcal{A} - M$; for $k = 0$ (i.e., the head and the body of a rule $r \in R$ are simultaneously absent) this condition does not hold, so no model exists;
3.  for rules $a \leftarrow b_1, \ldots, b_k.$ with non-empty head and $k > 0$, $b_1, \ldots, b_k \in M$ implies $a \in M$.

Alternatively, if all atoms were treated as Boolean variables (i.e., presence is true, absence is false), $M$ would be the model of an $R$ exactly when all rules (i.e., clauses) are satisfied.

The semantics of a program is defined by an answer set as follows. The *reduct* $\Pi^X$ of a program $\Pi$ relative to a set $X$ of atoms is defined by

$$\Pi^X = \{a \leftarrow b_1, \ldots, b_k. \mid a \leftarrow b_1, \ldots, b_k, \sim c_1, \ldots, \sim c_m. \in \Pi \text{ and } \{c_1, \ldots, c_m\} \cap X = \varnothing\}. \quad (4)$$

The $\subseteq$-smallest model of $\Pi^X$ is denoted by $\text{Cn}(\Pi^X)$. A set $X$ of atoms is an *answer set* of $\Pi$ if $X = \text{Cn}(\Pi^X)$.

For the sake of simplicity, AnsProlog programs are written using variables (by convention, variables start with uppercase letters). Such programs are then grounded, i.e., transformed to programs with no variables, by applying a Herbrand substitution. Note, however, that clever grounding discards rules that are redundant, i.e., that can never apply, because some atoms in their bodies have no possibility to be derived [19]. For example, the program:

$el(a) \leftarrow .$
$el(b) \leftarrow .$
$equal(L, L) \leftarrow el(L).$
$neq(L, Y) \leftarrow el(L), el(Y), \sim equal(L, Y).$

can be transformed to $\Pi$:

$el(a) \leftarrow .$
$el(b) \leftarrow .$
$equal(a, a) \leftarrow el(a).$
$equal(b, b) \leftarrow el(b).$
$neq(a, a) \leftarrow el(a), el(a), \sim equal(a, a).$
$neq(a, b) \leftarrow el(a), el(b), \sim equal(a, b).$
$neq(b, a) \leftarrow el(b), el(a), \sim equal(b, a).$
$neq(b, b) \leftarrow el(b), el(b), \sim equal(b, b).$

which has a single answer set: $X = \{equal(a, a), equal(b, b), el(a), el(b), neq(b, a), neq(a, b)\}$. A reduct $\Pi^X$ becomes:

$el(a) \leftarrow .$
$el(b) \leftarrow .$
$equal(a, a) \leftarrow el(a).$
$equal(b, b) \leftarrow el(b).$
$neq(a, b) \leftarrow el(a), el(b).$
$neq(b, a) \leftarrow el(b), el(a).$

Its minimal model $\text{Cn}(\Pi^X)$ is just $X$. In other words, a set $X$ of atoms is an answer set of a logic program $\Pi$ if: (i) $X$ is a classical model of $\Pi$ and (ii) all atoms in $X$ are justified by some rule in $\Pi$.

Recently, Answer Set Programming has emerged as a declarative problem-solving paradigm. This particular way of programming in AnsProlog is well-suited for modeling and solving problems that involve common sense reasoning. It has been fruitfully used in a range of applications.

Early ASP solvers used backtracking to find solutions. With the evolution of Boolean SAT solvers, several ASP solvers were built on top them. The approach taken by these solvers was to convert the ASP formula into SAT propositions, apply the SAT solver, and then convert the solutions back to ASP form. Newer systems, such as Clasp (which is a part of the Clingo solver, https://potassco.org/clasp/), take advantage of the conflict-driven algorithms inspired by SAT, without the complete conversion into a Boolean-logic form. These approaches improve the performance significantly, often by an order of magnitude, over earlier backtracking algorithms [21].

## 3. Proposed Encoding for the Induction of NFA

Our translation reduces NFA identification into an AnsProlog program. Suppose we are given a sample $S$ over an alphabet $\Sigma$, and a positive integer $k$. We want to find a $k$-state NFA $A = (\Sigma, \{q_0, q_1, \ldots, q_{k-1}\}, q_0, F, \delta)$ such that every $w \in S_+$ is recognized by accepting and every $w \in S_-$ is recognized by rejecting. The parameter $k$ can be regarded as the degree of data generalization. The smallest $k$, say $k_0$, for which our logic program has an answer set, will give the most general automaton. As $k$ increases, we obtain a set of nested languages, the largest for $k_0$ and the smallest for some $k_m \geq k_0$. Usually, the running time for $k > k_0$ is shorter than for $k_0$.

Let $\mathrm{Pref}(S)$ be the set of all prefixes of $S_+ \cup S_-$. The relationship between an automaton $A$ and a sample $S$ in terms of ASP is constrained as shown below in seven groups of rules. In rules (5)–(24) the following convention for naming variables is used: $P$ stands for a prefix, $N$ stands for a number (state index), $I$, $J$, and $M$ also represent state indexes, $C$ stands for a character (the element of alphabet), $W$ stands for word (which is also a prefix), $U$ represents another prefix.

1. We have the following domain specification, i.e., our AnsProlog facts.

$$q(i) \leftarrow . \quad \text{for all } i \in \{0, 1, \ldots, k-1\}. \tag{5}$$

$$symbol(a) \leftarrow . \quad \text{for all } a \in \Sigma. \tag{6}$$

$$prefix(p) \leftarrow . \quad \text{for all } p \in \mathrm{Pref}(S). \tag{7}$$

$$positive(s) \leftarrow . \quad \text{for all } s \in S_+. \tag{8}$$

$$negative(s) \leftarrow . \quad \text{for all } s \in S_-. \tag{9}$$

$$join(u, a, v) \leftarrow . \quad \text{for all } u, v \in \mathrm{Pref}(S) \text{ and } a \in \Sigma \text{ such that } ua = v. \tag{10}$$

Facts (5) and (6) define the set of states $Q$ and the input alphabet $\Sigma$, while facts (7)–(9) describe the input sample. In particular, they define the prefixes as well as words to be recognized by accepting and rejecting, respectively.

Finally, fact (10) defines the concatenation operation, which given prefix $u \in \mathrm{Pref}(S)$ and symbol $a \in \Sigma$ produces prefix $v \in \mathrm{Pref}(S)$.

2. The next rules ensure that in an automaton $A$ every prefix goes to at least one state and every state is final or not.

$$x(P, N) \leftarrow prefix(P), q(N), \sim not\_x(P, N). \tag{11}$$

$$not\_x(P, N) \leftarrow prefix(P), q(N), \sim x(P, N). \tag{12}$$

$$has\_state(P) \leftarrow prefix(P), q(N), x(P, N). \tag{13}$$

$$\leftarrow prefix(P), \sim has\_state(P). \tag{14}$$

$$final(N) \leftarrow q(N), \sim not\_final(N). \tag{15}$$

$$not\_final(N) \leftarrow q(N), \sim final(N). \tag{16}$$

Rules (11) and (12) describe the reachability of states $q \in Q$ by prefixes $p \in \mathrm{Pref}(S)$. State $q$ is *reachable* by prefix $p$ *iff* the prefix can be read by following a series of transitions from state $q_0$ to

state $q$ (this series of transitions builds a *path* for prefix $p$). The unreachable states are described by the default negation rule *not_x*. Clearly, for every prefix $p \in \text{Pref}(S)$ and every state $q \in Q$, either (11) or (12) holds. Here $P$ (a prefix) and $N$ (a number, state index) are variables, which means that during the grounding they will be substituted for, respectively, every $p \in \text{Pref}(S)$ because of the atom *prefix(P)* in the body of the rule and for every $i \in \{0, 1, \ldots, k-1\}$ because of the atom $q(N)$ in the body of the rule. Notice that for every $p \in \text{Pref}(S)$ we already have fact *prefix(p)* and for every $i \in \{0, 1, \ldots, k-1\}$ we already have fact $q(i)$, which are the sources of this substitution. Rules (13) and (14) declare that for every prefix $p \in \text{Pref}(S)$ there has to be some reachable state $q \in Q$. These rules follow from the fact that the members of sets $S_+$ and $S_-$ have to be recognized by accepting or rejecting, respectively. In other words, for each $w \in (S_+ \cup S_-)$ there has to be at least one path in the inferred NFA.

Finally, rules (15) and (16) ensure that each state $q \in Q$ is either accepting (final) or rejecting (not final). Such rules as the pair (15) and (16) are recommended in ASP textbooks to specify that each element either is/has something or is/has not (refer for example to Chapter 4 of Chitta Baral's [18]).

3. For encoding transitions we will use predicates *delta*.

$$delta(I,\ C,\ J) \leftarrow q(I), symbol(C), q(J), \sim not\_delta(I,\ C,\ J). \tag{17}$$

$$not\_delta(I,\ C,\ J) \leftarrow q(I), symbol(C), q(J), \sim delta(I,\ C,\ J). \tag{18}$$

Rule (17) says that if there exists a transition between a pair of states $q_i, q_j \in Q$, marked with a symbol $c \in \Sigma$ then $delta(I, C, J)$ is in the model. Otherwise, the default negation rule *not_delta* applies (rule (18)).

4. Without sacrificing the generality, we can assume that $q_0$ is the initial state.

$$\leftarrow \sim x(\varepsilon,\ 0). \tag{19}$$

$$\leftarrow x(\varepsilon,\ N), q(N), N \neq 0. \tag{20}$$

Rules (19) and (20) mean that only state $q_0$ is reachable by the empty word $\varepsilon$.

5. Every counter-example has to be recognized by rejecting.

$$\leftarrow q(N), x(W,\ N), final(N), negative(W). \tag{21}$$

Recall that for the headless rules at least one predicate present in the body of the rule cannot be satisfied. Hence, rule (21) means that there is no final state that is reachable by any word $w \in S_-$.

6. Every example has to be recognized by accepting. In this rule we used an extension syntax of ASP—a choice construction. Here, it means that the number of final states, $q_n$, for which $((q_0, W), q_n) \in \bar{\delta}$ cannot be equal to 0 for any example $w$.

$$\leftarrow positive(W), \{\ final(N) : q(N), x(W,\ N)\ \} = 0. \tag{22}$$

7. Finally, there are mutual constraints between $x$ and *delta* predicates.

$$x(W,\ M) \leftarrow q(I), q(M), join(U,\ C,\ W), x(U,\ I), delta(I,\ C,\ M). \tag{23}$$

$$\leftarrow join(U,\ C,\ W), q(N), x(W,\ N), \{\ delta(J,\ C,\ N) : q(J), x(U,\ J)\ \} = 0. \tag{24}$$

Rule (23) says that for some state $r$ that is reachable by word $w = uc$, there exists some state $q_i$ reachable by word $u$ and there is a transition between states $q_i$ and $q_m$ with symbol $c$.

Similarly, rule (24) says that if there is a word $w = uc$ leading to some state $q_i \in Q$, then the number of transitions with symbol $c$ outgoing from a state reachable by word $u$ cannot be zero.

**Example 1.** *Let us see an example. Suppose we are given $S_+ = \{abc, c\}$, $S_- = \{a, ab\}$, and $k = 2$. Rules (5) to (10) concretize into:*

*q(0) ←.    q(1) ←.*
*symbol(a) ←.    symbol(b) ←.    symbol(c) ←.*
*prefix(ε) ←.    prefix(a) ←.    prefix(ab) ←.    prefix(c) ←.    prefix(abc) ←.*
*positive(c) ←.    positive(abc) ←.*
*negative(a) ←.    negative(ab) ←.*
*join(ε, a, a) ←.    join(ε, c, c) ←.    join(a, b, ab) ←.    join(ab, c, abc) ←.*

*Rules (11) to (24) always remain unchanged. This program has an answer set $\{q(0), \ldots, delta(1, b, 1), final(0)\}$. In order to construct an associated NFA it is enough to take all final and delta predicates, which define, respectively, final states and transitions of the resultant automaton. So we have obtained an NFA depicted in Figure 1.*

Additionally, in Appendix A there is a description of how answer sets are determined. In Appendix B a larger illustration is given.

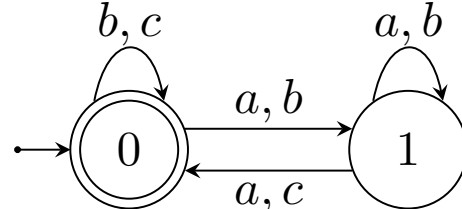

**Figure 1.** An inferred non-deterministic finite automaton (NFA).

## 4. Experimental Results

In this section, we describe some experiments comparing the performance of our approach (the program can be found at https://gitlab.com/wojtek3dan/asp4nfa) with the methods mentioned in the introductory section and described in more detail in Section 4.2. We used an ASP solver, Clingo, which can be executed sequentially or in parallel [22]. While comparing our approach with RA-PS1, RA-PS2, OA-PS1, and OA-PS2, all programs ran on an 8-core processor. ASP vs. SAT comparison was performed using a single core. For these experiments, we used a set of 40 samples (the samples can be found at https://gitlab.com/wojtek3dan/asp4nfa/-/tree/master/samples) based on randomly generated regular expressions.

*4.1. Benchmarks*

As far as we know, all standard benchmarks are too hard to be solved by pure exact algorithms. Thus, we generated problem instances using our own algorithm. This algorithm builds a set of words with the following parameters: size $|E|$ of a regular expression to be generated, alphabet size $|\Sigma|$, the number $|S|$ of words actually generated and their minimum, $d_{\min}$, and maximum, $d_{\max}$, lengths. The algorithm is arranged as follows. First, construct a random regular expression $E$. Next, obtain corresponding minimum-state DFA $M$. Then, as long as a sample $S$ is not symmetrically structurally complete (refer to Chapter 6 of [3] for the formal definition of this concept) with respect to $M$, repeat the following steps: (a) using the Xeger library (https://pypi.org/project/xeger/) for generating random strings from a regular expression, get two words $u$ and $w$; (b) truncate as few symbols from the end of $w$ as possible in order to obtain a counter-example $\bar{w}$; if it succeeds, add $u$ to $S_+$ and $\bar{w}$ to $S_-$. Finally, accept $S = (S_+, S_-)$ as a valid sample if it is not too small, too large or highly imbalanced. In order to ensure that these conditions are fulfilled, the equations $|S_+| \geq 8$, $|S_-| \geq 8$, and $|S_+| + |S_-| \leq 1000$ hold for all our samples. In generating a random word from a regex or from

an automaton we encounter a problem with, respectively, star operator and self-loops. Theoretically, there are infinitely many words matched to these fragments, so we have to bound the number of repetitions. We set this parameter to four.

In this manner we produced 40 sets with: $|E| \in [27, 46]$, $|\Sigma| \in \{2, 4, 6, 8\}$, $|S| \in [27, 958]$, $d_{\min} = 0$, and $d_{\max} = 305$. The file names with samples have the form 'a$|\Sigma|$words$|E|$.txt'. To give the reader a hint on the variability of the resulting automata, we show in Table 1 the numbers of states and transitions in each of the 40 NFAs found using our approach. We show there also the size of $M$ and the size of minimal DFA $D$ compatible with sample data. Example solutions from each group of problems, defined by the size of the input alphabet $|\Sigma|$ are also shown in Figure 2.

**Table 1.** Sizes of NFAs found by the answer set programming (ASP) solver ($k_0$—number of states, $t$—number of transitions, $|M|$—number of states in deterministic finite automaton (DFA) $M$, $|D|$—number of states in minimal DFA $D$ compatible with sample data).

| Problem | $k_0$ | $t$ | $\|M\|$ | $\|D\|$ | Problem | $k_0$ | $t$ | $\|M\|$ | $\|D\|$ |
|---|---|---|---|---|---|---|---|---|---|
| a2words27 | 8 | 32 | 20 | 15 | a6words27 | 3 | 23 | 9 | 3 |
| a2words28 | 3 | 6 | 7 | 3 | a6words28 | 4 | 50 | 15 | 4 |
| a2words29 | 5 | 22 | 10 | 6 | a6words29 | 3 | 24 | 6 | 3 |
| a2words30 | 3 | 7 | 6 | 3 | a6words30 | 2 | 14 | 8 | 2 |
| a2words31 | 6 | 13 | 12 | 7 | a6words31 | 2 | 19 | 11 | 2 |
| a2words32 | 5 | 10 | 10 | 5 | a6words32 | 2 | 11 | 6 | 2 |
| a2words33 | 7 | 19 | 17 | 8 | a6words33 | 7 | 97 | 17 | 10 |
| a2words34 | 4 | 13 | 6 | 5 | a6words34 | 5 | 58 | 16 | 6 |
| a2words35 | 3 | 6 | 6 | 3 | a6words35 | 2 | 14 | 6 | 2 |
| a2words36 | 5 | 23 | 9 | 8 | a6words36 | 4 | 41 | 15 | 5 |
| a4words27 | 4 | 32 | 10 | 4 | a8words37 | 2 | 19 | 8 | 3 |
| a4words28 | 3 | 16 | 14 | 3 | a8words38 | 3 | 34 | 6 | 3 |
| a4words29 | 5 | 34 | 6 | 6 | a8words39 | 2 | 27 | 11 | 2 |
| a4words30 | 3 | 15 | 7 | 4 | a8words40 | 3 | 30 | 13 | 3 |
| a4words31 | 4 | 23 | 12 | 4 | a8words41 | 5 | 79 | 11 | 6 |
| a4words32 | 3 | 18 | 6 | 3 | a8words42 | 4 | 65 | 21 | 5 |
| a4words33 | 4 | 24 | 12 | 4 | a8words43 | 4 | 60 | 15 | 4 |
| a4words34 | 4 | 21 | 16 | 4 | a8words44 | 2 | 23 | 7 | 2 |
| a4words35 | 4 | 28 | 23 | 4 | a8words45 | 5 | 66 | 15 | 7 |
| a4words36 | 2 | 9 | 5 | 3 | a8words46 | 3 | 27 | 9 | 3 |

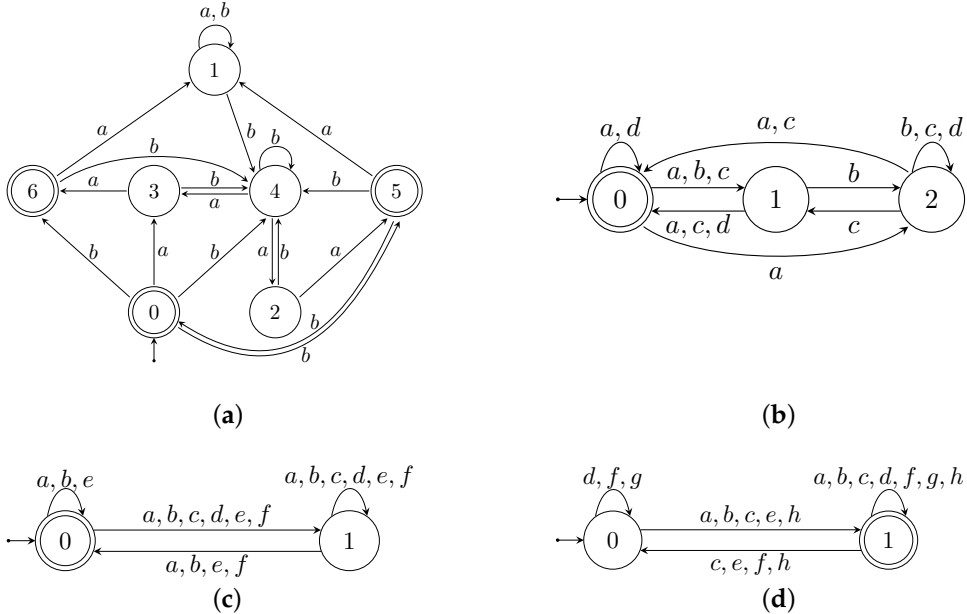

**Figure 2.** Example solutions found by the ASP solver for problems a2words33, a4words28, a6words31, and a8words37. (**a**) Problem a2words33; (**b**) Problem a4words28; (**c**) Problem a6words31; (**d**) Problem a8words37.

*4.2. Compared Algorithms*

As already mentioned, our algorithm was compared with a SAT-based algorithm and several exact parallel algorithms. To make the paper self-contained let us briefly describe these algorithms.

The SAT-based algorithm defines three types of binary variables, $x_{wq}$, $y_{apq}$, and $z_q$, for $w \in \mathrm{Pref}(S)$, $a \in \Sigma$, $p, q \in Q$. Variable $x_{wq} = 1$ *iff* state $q$ is reachable by prefix $w$, otherwise $x_{wq} = 0$. Variable $y_{apq} = 1$ *iff* there exists a transition from state $p$ to state $q$ with symbol $a$, otherwise $y_{apq} = 0$. Finally, $z_q = 1$ *iff* state $q$ is final, and $z_q = 0$ otherwise. The constraints involving these variables are as follows:

1.  All examples have to be accepted, while none of the counter-examples should be, which is described by

$$\forall_{w \in S_+ - \{\varepsilon\}} \sum_{q \in Q} x_{wq} z_q \geq 1, \tag{25}$$

$$\forall_{w \in S_- - \{\varepsilon\}} \sum_{q \in Q} x_{wq} z_q = 0. \tag{26}$$

2.  All prefixes $w = a$, $w \in \mathrm{Pref}(S)$, $a \in \Sigma$, result from the transitions outgoing from state $q_0$

$$\forall_{w=a} x_{wq} - y_{aq_0q} = 0. \tag{27}$$

3.  For all states $q \in Q$ reachable by prefixes $w = va$, $v, w \in \mathrm{Pref}(S)$, $a \in \Sigma$, there has to be some state $r$ reachable by prefix $v$, and there has to be an outgoing transition from $r$ to $q$ with symbol $a$. By symmetry, if there exists a path for prefix $v$ ending in some state $r$ and there exists a transition from $r$ to $q$ with symbol $a$ then there exists a path to state $q$ with prefix $w = va$. These conditions are expressed as

$$\forall_{w=va} - x_{wq} + \sum_{r \in Q} x_{vr} y_{arq} \geq 0, \tag{28}$$

$$\forall_{q,r \in Q} x_{wq} - x_{vr} y_{arq} \geq 0. \tag{29}$$

Additionally, it holds that $z_{q_0} = 1$, when $\varepsilon \in S_+$, $z_{q_0} = 0$, when $\varepsilon \in S_-$, and $z_{q_0}$ is not predefined when $\varepsilon \notin (S_+ \cup S_-)$. The solution to the presented problem formulation is sought by a SAT solver.

**Example 2.** *Let us consider Example 1 again. In the SAT-based formulation we have the following variables* $x_{aq_0}$, $x_{aq_1}$, ..., $x_{abcq_1}$, $y_{aq_0q_0}$, $y_{aq_0q_1}$, ..., $y_{cq_1q_1}$, $z_{q_0}$, *and* $z_{q_1}$. *Constraints (25)–(29) remain unchanged. A set of assignments satisfying the constraints at hand is as follows:* $x_{aq_1} = 1$, $x_{abq_1} = 1$, $x_{cq_0} = 1$, $x_{abcq_0} = 1$, $y_{aq_0q_1} = 1$, $y_{bq_1q_1} = 1$, $y_{cq_0q_0} = 1$, $y_{cq_1q_0} = 1$, $z_{q_0} = 1$. *All remaining variables are zeros. The resulting NFA is shown in Figure 3. Note that even though the set of transitions in Figure 3 is smaller than in Figure 1 both solutions are valid.*

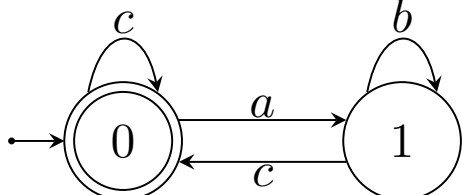

**Figure 3.** An inferred NFA.

Identification of a $k$-state NFA by means of the exact algorithms RA-PS1, RA-PS2, OA-PS1, and OA-PS2 is based on the SAT formulation given before. Assuming $k$ is fixed we only need to determine the set of final states $F$ and the transition function $\delta$. Let us recall that a set of final states is *feasible iff* the following conditions are satisfied: (i) $F \neq \varnothing$, (ii) $q_0 \in F$, if $\varepsilon \in S_+$, (iii) $q_0 \notin F$, if $\varepsilon \in S_-$. Clearly, an NFA without final states cannot accept any word, and if the empty word $\varepsilon$ is in $S_+$ (resp. $S_-$) the initial state $q_0$ has to be final (resp. not final). Since every feasible set $F$ may lead to an NFA consistent with the sample $S$ (as the NFAs need not be unique), we distribute the different sets $F$ among processes and try to identify the $\delta$ function by means of a backtracking algorithm.

While searching for the values of $y_{apq}$ variables, we apply different search orders. This is so, because there is no universal ordering method assuring fast convergence to the solution. The orderings used in the analyzed algorithms are *deg*, *mmex*, and *mmcex*. The *deg* ordering is a static ordering method based on a variable degree, i.e., the number of constraints the variables are involved in. The ordering does not change as the algorithm progresses. The *mmex* and *mmcex* orderings change dynamically while the algorithm runs. They aim at satisfying first the equations related to examples, or counter-examples, respectively.

The Parallelization Scheme 1 (PS1) maximizes the number of sets $F$ processed simultaneously. If the number of available processes is greater than the number of sets $F$ to be analyzed, we assign multiple variable orderings (VOs) to each set. In the RA-PS1 algorithm this assignment is performed randomly, while in the OA-PS1 algorithm, the *deg*, *mmex*, and *mmcex* methods are ordered by their complexity and chosen in a round robin fashion.

The Parallelization Scheme 2 (PS2) maximizes the number of variable orderings applied to the same set $F$. This way we shorten the time needed to obtain an answer whether an NFA exists for the given set $F$. If the number of available processes is smaller than the product of the number of sets $F$ and the number of variable orderings used, we need to choose the sets $F$ to be processed first. In the RA-PS2 algorithm we choose them at random, while in the OA-RS2 algorithm, we analyze first the sets for which the size of $F$ is smaller.

**Example 3.** *Let us consider the problem given in Example 1. Since $k = 2$ and $\varepsilon \notin (S_+ \cup S_-)$, the following sets $F$ can be defined:* $F_1 = \{q_0\}$, $F_2 = \{q_1\}$, *and* $F_3 = \{q_0, q_1\}$. *Let us also assume that we can use the three VOs discussed before. Finally, let the number of processes $p = 3$ (denoted by $p_i$, for $i = 0, 1, 2$). We can have the following example configurations of algorithms RA-PS1, RA-PS2, OA-PS1, OA-PS2:*

1. *Algorithm RA-PS1—process $p_0$ gets $(F_1, VO_3)$; process $p_1$ gets $(F_2, VO_2)$; process $p_2$ gets $(F_3, VO_3)$. Each process uses a single VO to analyze one of the possible sets $F_i$, $i = 1, 2, 3$. There is no guarantee that all VOs are used at least once.*

2. *Algorithm RA-PS2—process $p_0$ gets $(F_2, VO_1)$; process $p_1$ gets $(F_2, VO_2)$; process $p_2$ gets $(F_2, VO_3)$. Each process uses a different VO to analyze just one set F at a time (chosen randomly). If there is no solution for set $F_2$, we need to repeat the above assignments but this time for the set $F_1$ or $F_3$ (again chosen randomly). We repeat the above procedure until the solution is found.*

3. *Algorithm OA-PS1—process $p_0$ gets $(F_1, VO_1)$; process $p_1$ gets $(F_2, VO_2)$; process $p_2$ gets $(F_3, VO_3)$. Each process uses a single VO to analyze one of the possible sets $F_i$, $i = 1, 2, 3$, but this time the VOs are assigned according to a predefined order.*

4. *Algorithm OA-PS2—process $p_0$ gets $(F_1, VO_1)$; process $p_1$ gets $(F_1, VO_2)$; process $p_2$ gets $(F_1, VO_3)$. Each process uses a different VO to analyze just one set F at a time, but this time we start with $F_1$ followed by $F_2$ and $F_3$ (unless the solution is found at some stage).*

Note that in Parallelization Scheme 2, obtaining a negative answer, i.e., that an NFA does not exist for the given set $F_i$, by means of one VO allows us to stop the execution of other VOs and move on to another set $F_j$, $i \neq j$.

### 4.3. Performance Comparison

In all experiments, we used Intel (Santa Clara, California, U.S.) Xeon CPU E5-2650 v2, 2.6 GHz (8 cores, 16 threads), under Ubuntu 18.04 operating system with 190 GB RAM. The time limit (TL) was set to 1000 s. The results are listed in Table 2. In order to determine whether the observed mean difference between ASP and the remaining methods is a real CPU time decrease, we used a paired samples *t* test [23] pp. 1560–1565, for ASP vs. SAT, ASP vs. RA-PS1, ASP vs. RA-PS2, ASP vs. OA-PS1, and ASP vs. OA-PS2. As we can see from Table 3, *p* value is low in all cases, so we can conclude that our results did not occur by chance and that using our ASP encoding is likely to improve CPU time performance for prepared benchmarks.

Let us explain how the mean values were computed. All TL cells were substituted by 1000. Notice that this procedure does not violate the significance of the statistical tests, because our program completed computations within the time limit for all problems (files). Thus, determining all running times would even strengthen our hypothesis.

To make the advantage of the ASP-based approach over the exact parallel algorithms even more convincing let us analyze the largest sizes of automata analyzed by the algorithms within the time limit TL = 1000 s. The summary of obtained sizes is given in Table 4. Note that the table includes only the problems for which TL entries exist in Table 2. The entries marked with * denote executions in which the algorithms started running for the given *k* but were terminated due to the time limit, without producing the final NFA.

**Table 2.** Execution times of exact solving NFA identification in seconds.

| Problem | Execution | | | | | | |
| --- | --- | --- | --- | --- | --- | --- | --- |
| | Sequential | | Parallel | | | | |
| | SAT | ASP | RA-PS1 | RA-PS2 | OA-PS1 | OA-PS2 | ASP |
| a2words27 | 148.11 | 77.46 | TL | TL | TL | TL | 10.69 |
| a2words28 | 1.86 | 0.18 | 1.07 | 1.15 | 1.28 | 1.19 | 0.25 |
| a2words29 | 6.40 | 0.49 | TL | TL | TL | TL | 0.40 |
| a2words30 | 2.78 | 0.39 | 9.14 | 8.52 | 9.55 | 9.08 | 0.25 |
| a2words31 | 2.08 | 0.52 | TL | TL | TL | TL | 0.37 |
| a2words32 | 791.19 | 4.50 | TL | TL | TL | TL | 3.50 |
| a2words33 | 49.72 | 1.99 | TL | TL | TL | TL | 0.93 |
| a2words34 | 10.05 | 0.57 | 20.59 | 20.92 | 19.88 | 19.02 | 0.59 |
| a2words35 | 0.47 | 0.09 | 0.53 | 0.38 | 0.38 | 0.38 | 0.10 |
| a2words36 | 526.07 | 2.57 | TL | TL | TL | TL | 3.47 |

**Table 2.** *Cont.*

| Problem | Sequential | | Parallel | | | | |
| | SAT | ASP | RA-PS1 | RA-PS2 | OA-PS1 | OA-PS2 | ASP |
| --- | --- | --- | --- | --- | --- | --- | --- |
| a4words27 | 1.32 | 0.15 | TL | TL | TL | TL | 0.17 |
| a4words28 | 0.13 | 0.04 | 6.45 | 6.89 | 7.23 | 4.49 | 0.05 |
| a4words29 | 78.93 | 0.96 | TL | TL | TL | TL | 1.15 |
| a4words30 | 0.20 | 0.05 | 0.48 | 0.50 | 0.48 | 0.44 | 0.06 |
| a4words31 | 21.78 | 0.70 | TL | TL | TL | TL | 1.08 |
| a4words32 | 0.83 | 0.13 | 1.39 | 2.10 | 1.24 | 1.23 | 0.13 |
| a4words33 | 7.19 | 0.48 | 729.40 | 719.59 | 738.89 | 733.21 | 0.50 |
| a4words34 | 20.33 | 0.81 | TL | TL | TL | TL | 0.82 |
| a4words35 | 21.00 | 0.65 | TL | TL | TL | TL | 0.71 |
| a4words36 | 0.77 | 0.16 | 0.76 | 0.86 | 0.93 | 0.90 | 0.13 |
| a6words27 | 1.13 | 0.15 | 1.59 | 2.24 | 2.89 | 1.88 | 0.16 |
| a6words28 | 4.62 | 0.37 | TL | TL | TL | TL | 0.33 |
| a6words29 | 1.05 | 0.13 | 3.15 | 2.38 | 2.48 | 2.63 | 0.16 |
| a6words30 | 1.97 | 0.20 | 0.85 | 0.52 | 1.11 | 1.06 | 0.21 |
| a6words31 | 1.43 | 0.18 | 0.86 | 0.81 | 0.94 | 0.83 | 0.22 |
| a6words32 | 1.15 | 0.16 | 2.10 | 1.97 | 2.13 | 2.16 | 0.16 |
| a6words33 | 441.95 | 4.14 | TL | TL | TL | TL | 2.03 |
| a6words34 | 272.27 | 1.88 | TL | TL | TL | TL | 1.86 |
| a6words35 | 0.20 | 0.06 | 0.85 | 0.73 | 0.82 | 0.82 | 0.06 |
| a6words36 | 15.97 | 0.86 | TL | TL | TL | TL | 0.81 |
| a8words37 | 0.20 | 0.06 | 0.46 | 0.50 | 0.37 | 0.41 | 0.06 |
| a8words38 | 63.71 | 1.40 | 15.73 | 16.15 | 16.09 | 18.08 | 1.24 |
| a8words39 | 1.44 | 0.19 | 1.55 | 1.55 | 1.78 | 1.48 | 0.20 |
| a8words40 | 7.63 | 0.44 | TL | TL | TL | TL | 0.66 |
| a8words41 | 324.29 | 1.94 | TL | TL | TL | TL | 2.61 |
| a8words42 | 5.86 | 0.35 | TL | TL | TL | TL | 0.42 |
| a8words43 | 9.14 | 0.45 | TL | TL | TL | TL | 0.69 |
| a8words44 | 1.23 | 0.21 | 1.52 | 1.34 | 1.25 | 1.22 | 0.38 |
| a8words45 | 279.74 | 2.07 | TL | TL | TL | TL | 2.32 |
| a8words46 | 5.20 | 0.33 | TL | TL | TL | TL | 0.58 |
| Mean | 78.29 | 2.71 | 544.96 | 544.73 | 545.24 | 545.01 | 1.01 |

**Table 3.** Obtained *p* values from the paired samples *t* test.

| ASP vs. SAT | ASP vs. RA-PS1 | ASP vs. RA-PS2 | ASP vs. OA-PS1 | ASP vs. OA-PS2 |
| --- | --- | --- | --- | --- |
| 0.00773 | $2.72 \times 10^{-8}$ | $2.73 \times 10^{-8}$ | $2.70 \times 10^{-8}$ | $2.72 \times 10^{-8}$ |

**Table 4.** Sizes of NFAs reached by the parallel algorithms. The sign * means that the time limit was exceeded.

| Problem | ASP | RA-PS1 | RA-PS2 | OA-PS1 | OA-PS2 |
| --- | --- | --- | --- | --- | --- |
| a2words27 | 8 | 6 | 6 | 6 | 6 |
| a2words29 | 5 | 5 * | 5 * | 5 * | 5 * |
| a2words31 | 6 | 5 | 5 | 5 | 5 |
| a2words32 | 5 | 4 | 4 | 4 | 4 |
| a2words33 | 7 | 5 | 5 | 5 | 5 |
| a2words36 | 5 | 4 | 4 | 4 | 4 |
| a4words27 | 4 | 4 * | 4 * | 4 * | 4 * |
| a4words29 | 5 | 4 | 4 | 4 | 4 |
| a4words31 | 4 | 4 * | 4 * | 4 * | 4 * |
| a4words34 | 4 | 3 | 3 | 3 | 3 |
| a4words35 | 4 | 4 * | 4 * | 4 * | 4 * |

**Table 4.** *Cont.*

| Problem | ASP | RA-PS1 | RA-PS2 | OA-PS1 | OA-PS2 |
|---------|-----|--------|--------|--------|--------|
| a6words28 | 4 | 3 | 3 | 3 | 3 |
| a6words33 | 7 | 3 | 3 | 3 | 3 |
| a6words34 | 5 | 3 | 3 | 3 | 3 |
| a6words36 | 4 | 3 | 3 | 3 | 3 |
| a8words40 | 3 | 3 * | 3 * | 3 * | 3 * |
| a8words41 | 5 | 3 | 3 | 3 | 3 |
| a8words42 | 4 | 4 * | 4 * | 4 * | 4 * |
| a8words43 | 4 | 3 | 3 | 3 | 3 |
| a8words45 | 5 | 4 | 4 | 4 | 4 |
| a8words46 | 3 | 3 * | 3 * | 3 * | 3 * |

## 5. Conclusions

We have experimented with a model learning approach based on ASP solvers. The approach is very flexible, as proven by its successful adaptation for learning NFAs, implemented in the provided open source tool. Experiments indicate that our approach clearly outperforms the current state-of-the-art satisfiability-based method and all backtracking algorithms proposed in the literature. The approach does scale well (as far as non-deterministic acceptors are considered): we have shown that it can be used for learning models from up to a thousand words. In the future, we wish to develop more efficient encodings that will make the approach scale even better. We hope this paper encourages more interest in ASP-based problem solving since the presented approach has several benefits over traditional model learning algorithms. The ASP encoding is more readable than SAT encoding and the resulting program is much faster than its backtracking counterparts.

**Author Contributions:** Conceptualization, W.W.; methodology, O.U. and W.W.; software, T.J. and W.W.; validation, T.J. and O.U.; formal analysis, O.U.; investigation, W.W. and T.J.; resources, W.W.; writing—original draft preparation, W.W. and T.J.; writing—review and editing, O.U. and T.J.; supervision, O.U.; project administration, O.U.; funding acquisition, O.U. All authors have read and agreed to the published version of the manuscript.

**Funding:** This research was supported by the National Science Center (Poland), grant number 2016/21/B/ST6/02158.

**Conflicts of Interest:** The authors declare no conflict of interest.

## Appendix A. How Answer Sets Are Computed

Let $\Pi$ be a grounded program, and let $\mathcal{A}$ be a set of atoms occurring in $\Pi$. Assume that atom $z$ is not in $\mathcal{A}$. Observe that every grounded program $\Pi$ that has rules

$$\leftarrow b_1, \ldots, b_k, \sim c_1, \ldots, \sim c_m. \tag{A1}$$

with empty head, can be transformed to a program without such rules by inserting $z$ and $\sim z$ in this manner:

$$z \leftarrow b_1, \ldots, b_k, \sim c_1, \ldots, \sim c_m, \sim z. \tag{A2}$$

A grounded program without empty-headed rules will be called *normal*.

A rule $r$ of the form:

$$a \leftarrow b_1, \ldots, b_k. \tag{A3}$$

where there is no default negation in the body, and the head is not empty, will be called *positive*. A program that contains only positive rules will be called positive too. We will denote by head($r$) the

set $\{a\}$, and by body$(r)$ the set $\{b_1, \ldots, b_k\}$. Now, let us define how a positive program $P$ can act on a set of atoms $X \subseteq \mathcal{A}$ (here $\mathcal{A}$ is a set of atoms occurring in $P$):

$$X^P = \{\text{head}(r) \mid r \in P \text{ and body}(r) \subseteq X\}. \tag{A4}$$

This operation can be repeated and we define:

$$X^{P^1} = X^P \quad \text{and} \quad X^{P^i} = (X^{P^{i-1}})^P. \tag{A5}$$

It is easy to see that $\text{Cn}(P) = \bigcup_{i \geq 1} \varnothing^{P^i}$. Because for a certain $i$ the equation $X^{P^i} = X^{P^{i+1}}$ holds, determining $\text{Cn}(P)$ is straightforward and fast.

Consider any normal program $\Pi$. We recall from Section 2.3 that a set $X \subseteq \mathcal{A}$ is an answer set of $\Pi$ if $X = \text{Cn}(\Pi^X)$ (please do not confuse the reduct with program's acting on a set of atoms). Take two sets, $L$ and $U$, such that $L \subseteq X \subseteq U$ for an answer set $X$ of $\Pi$. Observe that: (i) $X \subseteq \text{Cn}(\Pi^L)$, and (ii) $\text{Cn}(\Pi^U) \subseteq X$. Thus we get:

$$L \cup \text{Cn}(\Pi^U) \subseteq X \subseteq U \cap \text{Cn}(\Pi^L). \tag{A6}$$

The last property is a recipe for expanding the lower bound $L$ and cutting down the upper bound $U$. The procedure in which we replace $L$ by $L \cup \text{Cn}(\Pi^U)$ and then $U$ by $U \cap \text{Cn}(\Pi^L)$ as long as $L$ or $U$ are changed, will be called *narrowing*. At some point we get $L = U = X$. When we start from $L = \varnothing$, $U = \mathcal{A}$, then there are also two more possibilities: $L \not\subseteq U$ (there is no answer set), and $L \subset U$. In the latter case we can take any $a \in U - L$ and check out two paths: $a$ should be included into $L$ or $a$ should be excluded from $U$. This leads to Algorithm A1 [19]:

---
**Algorithm A1:** Final algorithm

---
$\textsc{Solve}(\Pi, L, U)$
    $(L, U) \leftarrow \text{narrowing}(\Pi, L, U)$
    **if** $L \not\subseteq U$ **then** return
    **if** $L = U$ **then** output $L$
    **else**
        choose $a \in U - L$
        $\textsc{Solve}(\Pi, L \cup \{a\}, U)$
        $\textsc{Solve}(\Pi, L, U - \{a\})$

---

Which outputs all answer sets of a program $\Pi$ provided that it had been invoked with $\textsc{Solve}(\Pi, \varnothing, \mathcal{A})$. The pessimistic time complexity of this algorithm can be assessed by the recurrence relation $T(n) = 2T(n-1) + n^2$, where $n = |U| - |L|$, which gives us the exponential complexity $T(n) = O(2^n)$.

## Appendix B. The Complete Example of an ASP Program for NFA Induction

Suppose we are given $\Sigma = \{a, b\}$, $S_+ = \{a\}$, $S_- = \{b\}$, and $k = 2$. After grounding rules (5)–(24) we get a program $\Pi$ in a Clasp format (symbol `:-` denotes left arrow, symbol `not` denotes default negation, and symbol `lambda` denotes the empty word $\varepsilon$):

```
symbol(a).
symbol(b).
prefix(lambda).
prefix(a).
prefix(b).
positive(a).
negative(b).
join(lambda,a,a).
```

```
join(lambda,b,b).
q(0).
q(1).
delta(0,a,0):-not not_delta(0,a,0).
delta(1,a,0):-not not_delta(1,a,0).
delta(0,b,0):-not not_delta(0,b,0).
delta(1,b,0):-not not_delta(1,b,0).
delta(0,a,1):-not not_delta(0,a,1).
delta(1,a,1):-not not_delta(1,a,1).
delta(0,b,1):-not not_delta(0,b,1).
delta(1,b,1):-not not_delta(1,b,1).
not_delta(0,a,0):-not delta(0,a,0).
not_delta(1,a,0):-not delta(1,a,0).
not_delta(0,b,0):-not delta(0,b,0).
not_delta(1,b,0):-not delta(1,b,0).
not_delta(0,a,1):-not delta(0,a,1).
not_delta(1,a,1):-not delta(1,a,1).
not_delta(0,b,1):-not delta(0,b,1).
not_delta(1,b,1):-not delta(1,b,1).
x(lambda,0):-not not_x(lambda,0).
x(a,0):-not not_x(a,0).
x(b,0):-not not_x(b,0).
x(lambda,1):-not not_x(lambda,1).
x(a,1):-not not_x(a,1).
x(b,1):-not not_x(b,1).
not_x(lambda,0):-not x(lambda,0).
not_x(a,0):-not x(a,0).
not_x(b,0):-not x(b,0).
not_x(lambda,1):-not x(lambda,1).
not_x(a,1):-not x(a,1).
not_x(b,1):-not x(b,1).
x(a,0):-delta(1,a,0),x(lambda,1).
x(a,1):-delta(1,a,1),x(lambda,1).
x(b,0):-delta(1,b,0),x(lambda,1).
x(b,1):-delta(1,b,1),x(lambda,1).
x(a,0):-delta(0,a,0),x(lambda,0).
x(a,1):-delta(0,a,1),x(lambda,0).
x(b,0):-delta(0,b,0),x(lambda,0).
x(b,1):-delta(0,b,1),x(lambda,0).
:-x(a,0),0>=#count{0,delta(0,a,0):x(lambda,0),delta(0,a,0);
0,delta(1,a,0):delta(1,a,0),x(lambda,1)}.
:-x(a,1),0>=#count{0,delta(0,a,1):x(lambda,0),delta(0,a,1);
0,delta(1,a,1):x(lambda,1),delta(1,a,1)}.
:-x(b,0),0>=#count{0,delta(0,b,0):x(lambda,0),delta(0,b,0);
0,delta(1,b,0):x(lambda,1),delta(1,b,0)}.
:-x(b,1),0>=#count{0,delta(0,b,1):x(lambda,0),delta(0,b,1);
0,delta(1,b,1):x(lambda,1),delta(1,b,1)}.
final(0):-not not_final(0).
final(1):-not not_final(1).
not_final(0):-not final(0).
not_final(1):-not final(1).
:-0>=#count{0,final(0):final(0),x(a,0);0,final(1):final(1),x(a,1)}.
:-final(0),x(b,0).
:-final(1),x(b,1).
:-x(lambda,1).
```

```
:-not x(lambda,0).
has_state(lambda):-x(lambda,0).
has_state(a):-x(a,0).
has_state(b):-x(b,0).
has_state(lambda):-x(lambda,1).
has_state(a):-x(a,1).
has_state(b):-x(b,1).
:-not has_state(lambda).
:-not has_state(a).
:-not has_state(b).
```

One of the answer sets $X$ is:

```
q(0) q(1) prefix(lambda) prefix(a) prefix(b) symbol(a) symbol(b)
negative(b) positive(a) join(lambda,a,a) join(lambda,b,b) x(lambda,0)
not_x(lambda,1) has_state(lambda) not_delta(0,a,0) not_delta(1,a,0)
delta(0,b,0) not_delta(1,b,0) delta(0,a,1) not_delta(1,a,1)
not_delta(0,b,1) not_delta(1,b,1) not_x(a,0) x(b,0) x(a,1)
not_x(b,1) not_final(0) final(1) has_state(a) has_state(b)
```

Note that the above answer set corresponds to a 2-state automaton having non-final state $q_0$ and final state $q_1$ (see predicates `not_final(0)` and `final(1)`), and the following transitions $q_0 \in \delta(q_0, b)$, $q_1 \in \delta(q_0, a)$ (defined by predicates `delta(0,b,0)` and `delta(0,a,1)`).

It can be easily verified that $\Pi^X = P$ is the positive program:

```
symbol(a).
symbol(b).
prefix(lambda).
prefix(a).
prefix(b).
positive(a).
negative(b).
join(lambda,a,a).
join(lambda,b,b).
q(0).
q(1).
delta(0,b,0).
delta(0,a,1).
not_delta(0,a,0).
not_delta(1,a,0).
not_delta(1,b,0).
not_delta(1,a,1).
not_delta(0,b,1).
not_delta(1,b,1).
x(lambda,0).
x(b,0).
x(a,1).
not_x(a,0).
not_x(lambda,1).
not_x(b,1).
x(a,0):-delta(1,a,0),x(lambda,1).
x(a,1):-delta(1,a,1),x(lambda,1).
x(b,0):-delta(1,b,0),x(lambda,1).
x(b,1):-delta(1,b,1),x(lambda,1).
x(a,0):-delta(0,a,0),x(lambda,0).
x(a,1):-delta(0,a,1),x(lambda,0).
```

```
x(b,0):-delta(0,b,0),x(lambda,0).
x(b,1):-delta(0,b,1),x(lambda,0).
final(1).
not_final(0).
has_state(lambda):-x(lambda,0).
has_state(a):-x(a,0).
has_state(b):-x(b,0).
has_state(lambda):-x(lambda,1).
has_state(a):-x(a,1).
has_state(b):-x(b,1).
```

$\mathrm{Cn}(P) = X$ since $\bigcup_{i \geq 1} \varnothing^{Pi} = \varnothing^P \cup (\varnothing^P)^P = X$. Further action of $P$ on $X$ does not change it.

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
