# Peer review of "Answer Set Programming for Regular Inference"

_applsci, doi:10.3390/app10217700_

Round 1

Reviewer 1 Report

The article discusses a sensible new solution for the induction of NFAs by using logic programming. The results and conclusion are believable and make sense. I think the article should get published after a few minor additions.

I recommend that they discuss their results more with explicit wordings on how they different tested solutions compare against themselves and against the benchmark DFAs in terms of size and correctness. I recommend that they provide an appendix that gives a larger illustration of how their ASP system works to induce an NFA.

My suggestions for improvement are elaborated more in the attached pdf (as anonymized comments).

Reviewer 2 Report

In this paper, the authors present a novel approach to NFA inference using answer set programming solvers. To that end, they come up with encoding rules to transform an input sample, as well as the size of the expected NFA into a logic program that the solver can resolve. Later, the answer set for the program is used to return an NFA consistent with the input sample.

The contributions of the paper are the following:
1. an NFA learning algorithm using ASP solvers, as well as
2. the implementation of the method, available on Github (which is a positive feature of the paper)

The results on a synthetic benchmark show that the method, compared to other methods, the one presented scales better.

I have several points of criticism about the paper:
- What is, in practice, the interest of setting a 'k', that is to (upper/lower)bound the number of states of the NFA? Could you imagine a scenario where 'k' is unknown? To what extent your algorithm can handle this uncertainty?
- There seems to be no proper literature review, which I think would have helped the reader put the paper in the context. Some parts of the paper about references could be expanded (for instance, I would have liked to see some details about the approaches of [11] described l.52 to make the section self-contained), and why not event mention some related work in grammatical inference using SAT solvers (for instance [1] even though concerning DFAs).
- l.74, are you sure of the term "catenation"?
- l.107, what do you actually mean by "the model of an R is a subset..."? If you refer to "set of rules", I would suggest the written down version rather than "R" in plain sentences.
- l.119, you say "by convention, variables start with uppercase letters", which you seem to stick to when you write rules in (11), (12), ... , (24). However, this leads to obvious conflicting notations already defined in the paper: for instance, in (17) and (18), A and R are already defined in lines 82 and 106 as defining an NFA and a set of rules, respectively.
- Similarly, in (11) and (12), to what P and N refer? I would suggest rewriting Sect. 3 and give thorough explanations.
- Concerning your implementation, I have 2 concerns: 1. I doubt whether you use all of the rules defined from (5) to (24), and 2. the Github repository deserves a Readme (packages to install, using FAdo port for Python3, etc.)
- In Sect. 4.1 you again have conflicting notations (l.212 with that time A referring to alphabet size)
- l. 291, you literally say "We owe you an explanation of computing the mean values", which is an "interesting" phrasing (there are also other ones like l.264 "once again" ...)

Despite these, I think the contributions are sound and relevant and are of good interest to the research community.

References
[1] Smetsers, R., Fiterău-Broştean, P., & Vaandrager, F. (2018, April). Model learning as a satisfiability modulo theories problem. In International Conference on Language and Automata Theory and Applications (pp. 182-194). Springer, Cham.
